# The optimal spatially-dependent control measures to effectively and economically eliminate emerging infectious diseases

**Fan Xia[1], Yanni Xiao[1], Junling Ma** [2]*

**1** School of Mathematics and Statistics, Xi'an Jiaotong University, Xi'an, China, **2** Department of Mathematics and Statistics, University of Victoria, Victoria, Canada

* junlingm@uvic.ca

**Data Availability Statement:** All data and codes used in the paper can be downloaded from https://github.com/nidefanwang/optimal_spatially_dependent_control_measures_for_emerging_infectious_diseases.git.

## Abstract

Non-pharmaceutical interventions (NPIs) are effective in mitigating infections during the early stages of an infectious disease outbreak. However, these measures incur significant economic and livelihood costs. To address this, we developed an optimal control framework aimed at identifying strategies that minimize such costs while ensuring full control of a cross-regional outbreak of emerging infectious diseases. Our approach uses a spatial SEIR model with interventions for the epidemic process, and incorporates population flow in a gravity model dependent on gross domestic product (GDP) and geographical distance. We applied this framework to identify an optimal control strategy for the COVID-19 outbreak caused by the Delta variant in Xi'an City, Shaanxi, China, between December 2021 and January 2022. The model was parameterized by fitting it to daily case data from each district of Xi'an City. Our findings indicate that an increase in the basic reproduction number, the latent period or the infectious period leads to a prolonged outbreak and a larger final size. This indicates that diseases with greater transmissibility are more challenging and costly to control, and so it is important for governments to quickly identify cases and implement control strategies. Indeed, the optimal control strategy we identified suggests that more costly control measures should be implemented as soon as they are deemed necessary. Our results demonstrate that optimal control regimes exhibit spatial, economic, and population heterogeneity. More populated and economically developed regions require a robust regular surveillance mechanism to ensure timely detection and control of imported infections. Regions with higher GDP tend to experience larger-scale epidemics and, consequently, require higher control costs. Notably, our proposed optimal strategy significantly reduced costs compared to the actual expenditures for the Xi'an outbreak.

## Author summary

In the early stage of the outbreak of an emerging infectious disease, non-pharmaceutical interventions (NPIs) are the most effective way to control the spread of the disease given unavailability of effective medicines or vaccines. However, the implementation of NPIs

**Funding:** This research is funded by Major International (Regional) Joint Research Project of National Natural Science Foundation of China (12220101001) (YX), National Key R&D Program of China (2022YFA1003704, YX), National Natural Science Foundation of China (12071366, FX), Chinese Scholarship Council (CSC, FX), Natural Sciences and Engineering Research Council of Canada (NSERC) Discovery Grant (JM), NSERC Emerging Infectious Disease Modeling Grants (MfPH and OMNI) (JM). The funders had no role in study design, data collection and analysis, decision to publish, or preparation of the manuscript.

may have a certain negative impact on the society and economy. There are many challenges to designing an effective and economical control regime on the basis of spatial, populate and economical heterogeneity. We developed a modelling framework to couple transmission dynamics, all possible control measures and the spatial mobility, aiming at controlling the outbreak spreading across regions and reducing the economic losses caused by control measures. We applied this framework to identify the optimal control strategy for the COVID-19 outbreak in Xi'an City, China, between December 2021 and January 2022. We analyzed the key factors related to the outbreak size and control cost, and the results showed that the outbreak size and control cost were significantly correlated with the transmissibility of the virus and the demographic and economic conditions of the epidemic area. Our results demonstrate that optimal control regimes exhibit spatial, economic, and population heterogeneity.

## Introduction

The World Health Statistics Report issued by the World Health Organization (WHO) shows that infectious diseases have been one of the greatest threats to human health since 2000, accounting for the second highest number of deaths among all mortality factors. Even before the start of coronavirus disease (COVID-19) pandemic in 2019, more than 10 million people died from infectious diseases worldwide [1]. Effective prevention and control of infectious diseases is therefore vital to human life and health. When an outbreak of an emerging infectious disease occurs, it is difficult to develop herd immunity through vaccination in a short period of time, and non-pharmaceutical interventions (NPIs) are the most effective way to control the spread of the disease. Taking COVID-19 for example, although there are differences in approaches to the prevention and control of epidemic, most countries have adopted strict non-pharmaceutical interventions, including lockdown, border closure and massive nucleic acid screening in the first wave of the outbreak [2–4].

Despite NPIs being able to effectively control the epidemic in a short period of time, their effects are achieved by limiting the gathering and movement of people. Large-scale NPIs restrict commercial activities, transportation and logistics, and enterprise production, thus causing a significant burden on normal social life and economic development [5–7]. These negative impacts of NPIs vary in severity across regions. The spread of an infectious disease is often influenced by various environmental and social factors [8, 9]. Regional differences in population, economy, and climate can lead to differences in the transmissibility of the disease in different regions, and such regional differences prompt local governments to adopt epidemic control policies of different intensity in response to the epidemic, which leads to distinction in the degree of economic loss caused by epidemic [10]. Due to such regional heterogeneity in disease transmission, policy makers need to consider how to formulate specific epidemic control policies based on the specific conditions of regions and minimize the negative impact of epidemic control measures on society and economy.

Assessing the efficacy of the implemented control measures is crucial for public health agencies for pandemic planning. Two approaches are commonly for the assessment. One is using case-control studies, the other is using mathematical modeling. Controlled studies have long been used to study the efficacy of vaccines (see, e.g., [11, 12]). This method can also be used to evaluate the reduction of infection risks from other control measures, such as wearing face masks [13] and social distancing [14]. However, these efficacy studies are pathogen specific, and may not be applicable to emerging pathogens or even new variants. Yet during an

emergency public health response to a pandemic, it may not be possible to conduct these case-control studies before implementing control policies. In addition, case-control studies are not suitable to study control measures targeting the identification of patients, such as contact tracing and serological screening. Thus, mathematical modeling has become an important tool to study the effectiveness fo control measures.

Mathematical modeling studies cover the research and evaluation of control effects of NPIs such as contact tracing [15–17], lockdown and social distancing [2, 4, 18–20], and travel restrictions and border control [21–26]. A wide range of mathematical tools has been used to model disease outbreaks and control measures, including probability and statistics [27–29], stochastic process [17, 30], and differential equations [2, 18, 20, 31].

Mathematical models have commonly been used to explore alternative policies to identify the most effective (or efficient) policy. This approach has been used during the COVID-19 pandemic to study vaccination strategies [32, 33], lockdowns [34], social distancing [35], school closure [36], and serological testing strategies [37].

Realistically, a combination of these policies are commonly implemented for disease control. Some policies, such as lockdown and travel restrictions, have large socio-economic costs, while some (e.g., contact tracing) are cost effective only when the scale of epidemic is not too large. To determine an optimal combination to achieve disease control with the most least socio-economic costs, the optimal control theory is typically used in combination with mathematical models of disease dynamics [38]. It finds the optimal control scheme under a given dynamic system. In the field of epidemic control, the optimal control theory is often used to solve problems including vaccine or medication allocation [39–41], testing strategy [42, 43], and deployment of control resources [43, 44]. Some consider a combination of control measures, such as social distancing and testing [45], and lockdown and treatment [46].

However, there lacks a modeling framework to study optimal disease control strategies that consider integrate major interventions of various effectiveness and costs at different phases of an epidemic, such as contact tracing, testing, travel restrictions, and lockdown of local municipalities. We propose a more integrated modeling approach and then build an optimal control problem that incorporates realistic epidemic, socio-economic and demographic factors. We hope to answer how to quantify epidemic control costs and reduce them under the premise of effective epidemic control, and explore the relevant factors of regional heterogeneity in epidemic transmission and control.

As an retrospective study, we calibrate the model to the COVID-19 outbreak occurring between December 2021 and January 2022 in Xi'an City, Shaanxi Province. To curtail this epidemic, the city implemented a sequence of control measures, starting with quarantine and isolation paired with contact tracing (a.k.a. precision control measures), district border closure, and blanket testing and city lockdown. Using this example, we hope to justify this framework as a new mathematical tool for effective regional policy recommendations that can be used for future emerging and resurgent infectious disease outbreaks.

## Methods

### Data

We use the COVID-19 outbreak in Xi'an City, Shaanxi Province, China between December 2021 and January 2022 as an example to illustrate our framework on optimal disease intervention strategies. Geographic information of Xi'an City is shown in Fig 1. The disease outbreak data is collected from Health Commission of Shaanxi Province and People's Government of Shaanxi Province [47, 48], including the daily number of new isolated cases and daily implementation of control measures in each district and county, as shown in Figs 2 and 3. The index

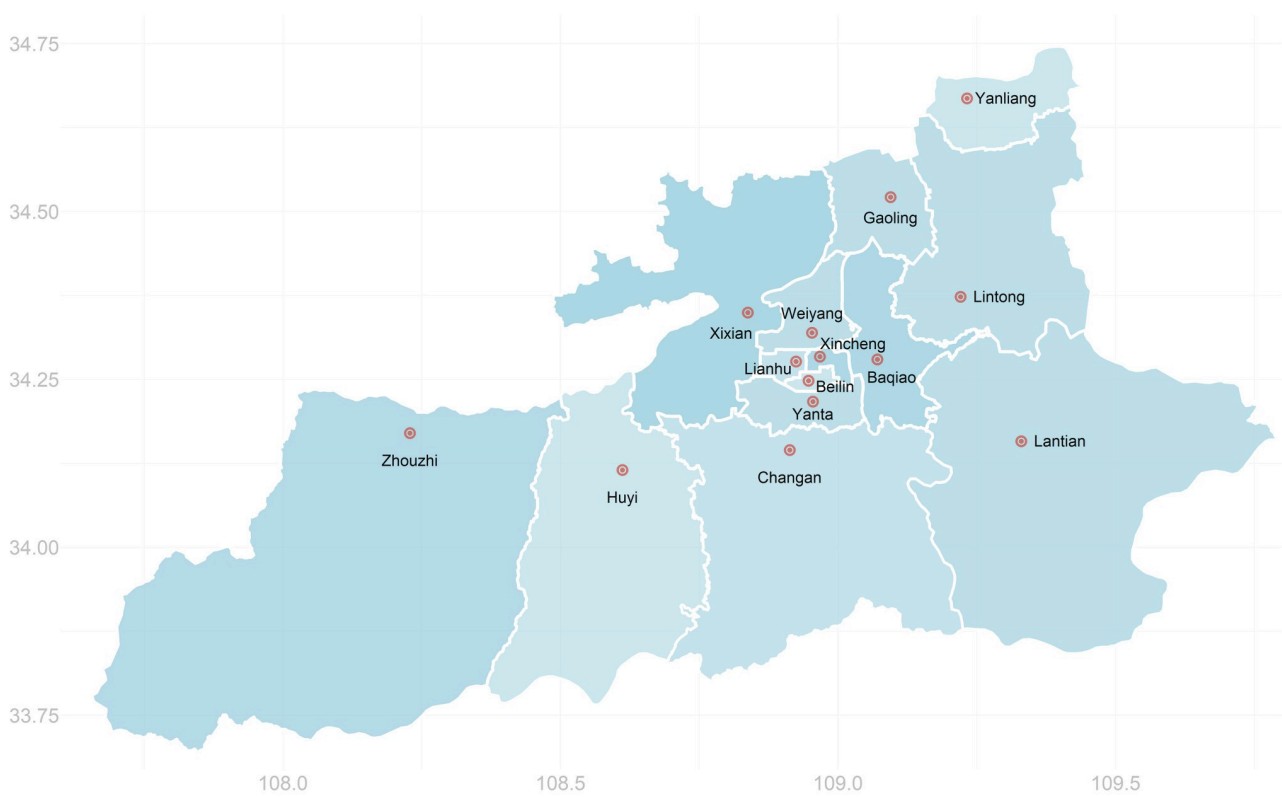

**Fig 1. Geographical locations of 14 districts and counties in Xi'an City, Shaanxi province, China.** The orange dots are the locations of the regional government buildings. The map data is extracted from https://openstreetmap.org (published under the Open Database 1.0 license), and is created using the ggplot2 package of R.

case of this outbreak was an imported case who entered Xi'an City through Xi'an Xianyang International Airport on December 4, 2021 [49]. The outbreak first started at Chang'an University in Yanta District, Xi'an City and spread to several surrounding Districts and counties. The first confirmed case at Chang'an University went to Xi'an Xianyang International Airport on December 4, 2021 and can be considered as the index case for subsequent community transmission of the outbreak [48]. No new case had been reported since January 20, 2022, and sequencing results showed that all cases were infected with the Delta variant (lineage B.1.617.2).

Population movement in the city drives the spatial spread within the city. Geographical adjacency is a key factor that affects this population flow. Xi'an City consists of 14 regions, namely Yanta District, Chang'an District, Lianhu District, Beilin District, Weiyang District, Baqiao District, Xincheng District, Yanliang District, Huyi District, Lintong District, Gaoling District, Zhouzhi County, Lantian County and Xixian New Area. Geographical adjacency relations of these regions are shown in Fig 1. Other key factors affecting population flow include demographic and economic status of the regions [50]. We use the demographic and economic statistics for Xi'an city in 2020 as an approximation for these factors during the outbreak. The data are obtained from Xi'an Municipal Bureau of Statistics [51]. shown in Fig 4. The data are plotted in Fig 4, which shows that Yanta District has the highest gross domestic product (GDP) of over 250 billion yuan in Xi'an. Combining Figs 1 and 4 we can see that economically developed districts such as Yanta, Lianhu and Beilin are located in the center of Xi'an City, moreover, these areas have a larger population size and are more densely-populated.

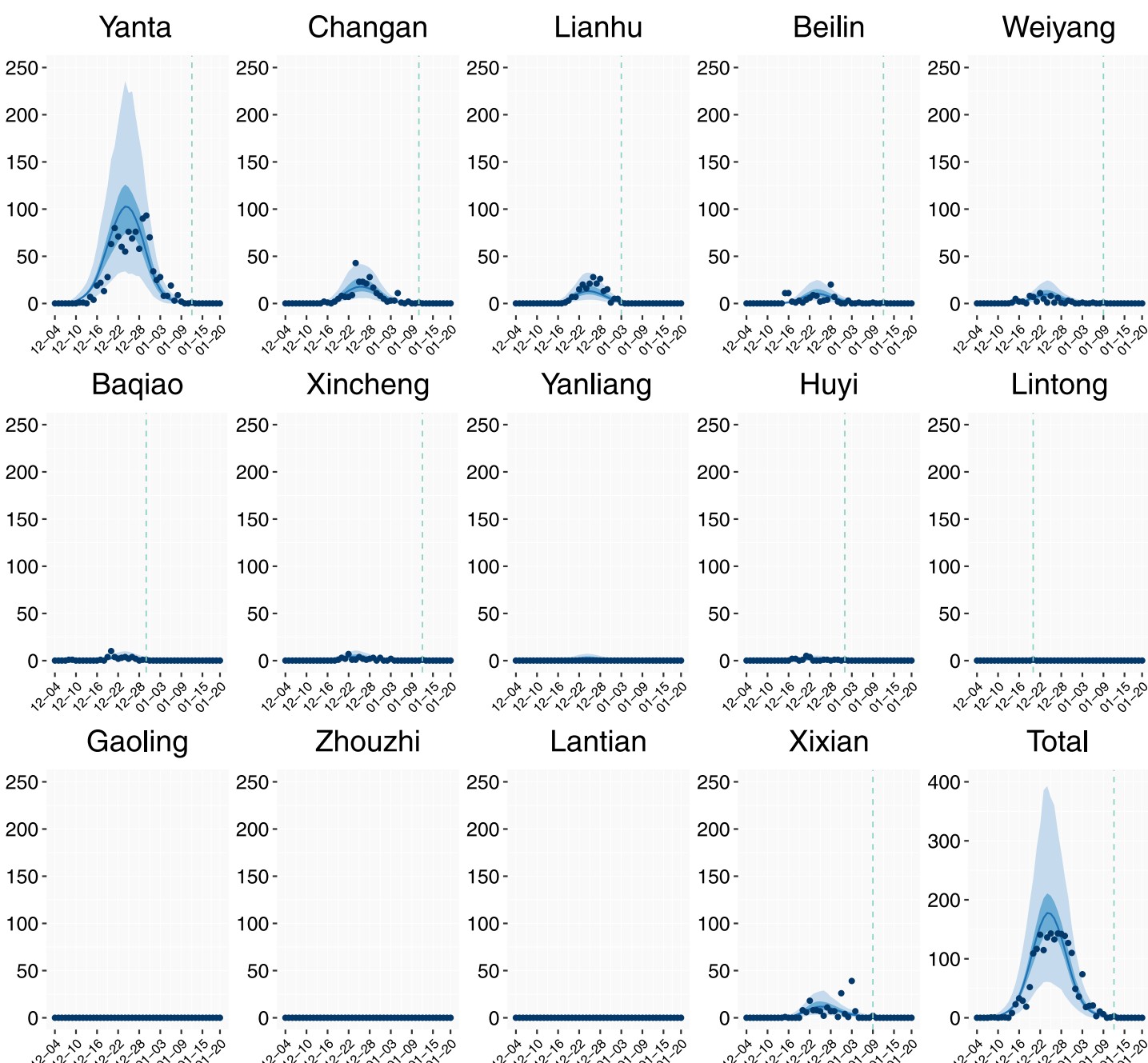

**Fig 2. Actual observations $\{x_i(t)\}_{i,t}$ (blue dots) and smoothed model fitting values $\mu_i(t; \hat{\theta})$ (blue lines) of daily new isolated cases from December 4, 2021 to January 20, 2022.** Dark blue shaded area is the smoothed 95% credible interval of $\mu_i(t; \hat{\theta})$. Light blue shaded area is the smoothed 95% credible interval for the posterior predictive distribution [64] of the number of new isolated cases. The vertical dashed line indicates the latest time of the appearance of new isolated cases, namely $t_{i,d}^{zero} = \max\{t : x_i(t) > 0\}$.

## Modeling the epidemic

In this subsection, we extend an multi-patch SEIR model to incorporate the spatial spread across the districts of Xi'an and various disease control measures implemented by the city. We divide the population into the Susceptible (*S*), Exposed (i.e., latent, *E*), Inectious (*I*) and

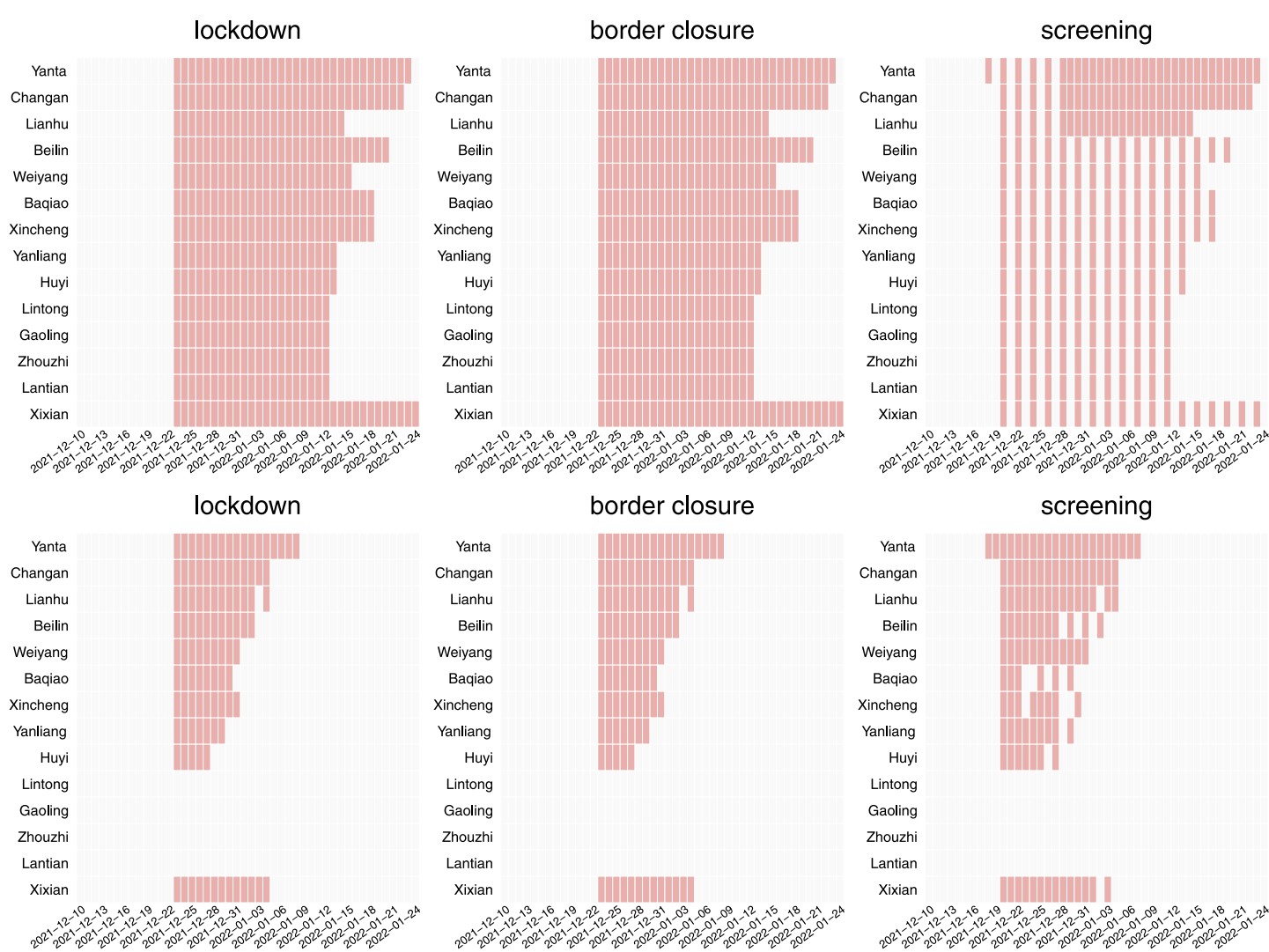

**Fig 3. Implementation scheme of control measures.** Top panel: Actual implementation scheme of lockdown, regional border closure, and universal nucleic acid screening during the epidemic in Xi'an City from December 10, 2021 ($t = 5$) when the first case was confirmed to January 24, 2022 ($t = 50$) when all control measures were lifted. Precise control in all regions began on December 10, 2021 and continued until the end of the outbreak. Bottom panel: Optimal implementation scheme control measures during the same period. In all subfigures, Rows represent regions, columns represent time, red and white represent implementation and non-implementation respectively.

Recovered ($R$) compartments that can freely contact other individuals, a quarantined compartment of exposed individuals from contact tracing $C^E$ and an isolation compartment of infectious individuals $C^I$ from diagnosis, and a hospitalized compartment of diagnosed patients ($H$). The individuals in $C^E$, $C^I$ and $H$ compartments are assumed to be fully isolated and cannot transmit. In addition. Each compartment is labeled with a subscript $i = 1, 2, \ldots, 14$ to denote the population in a district in Xi'an. Furthermore, the $E$ and $C^E$ compartments are also labeled by a superscript $\omega = m, s$ to denote whether these individuals will develop mild ($m$) or severe ($s$) symptoms, and the individuals in $I$ and $C^I$ compartments are also labeled by the superscript $\omega$ to denote whether they show severe symptoms. We assume that exposed individuals will develop severe symptoms with probability $q^s$ and mild symptoms with probability $q^m = 1 - q^s$, and severe cases will be hospitalized before recovery. The total exposed and infectious

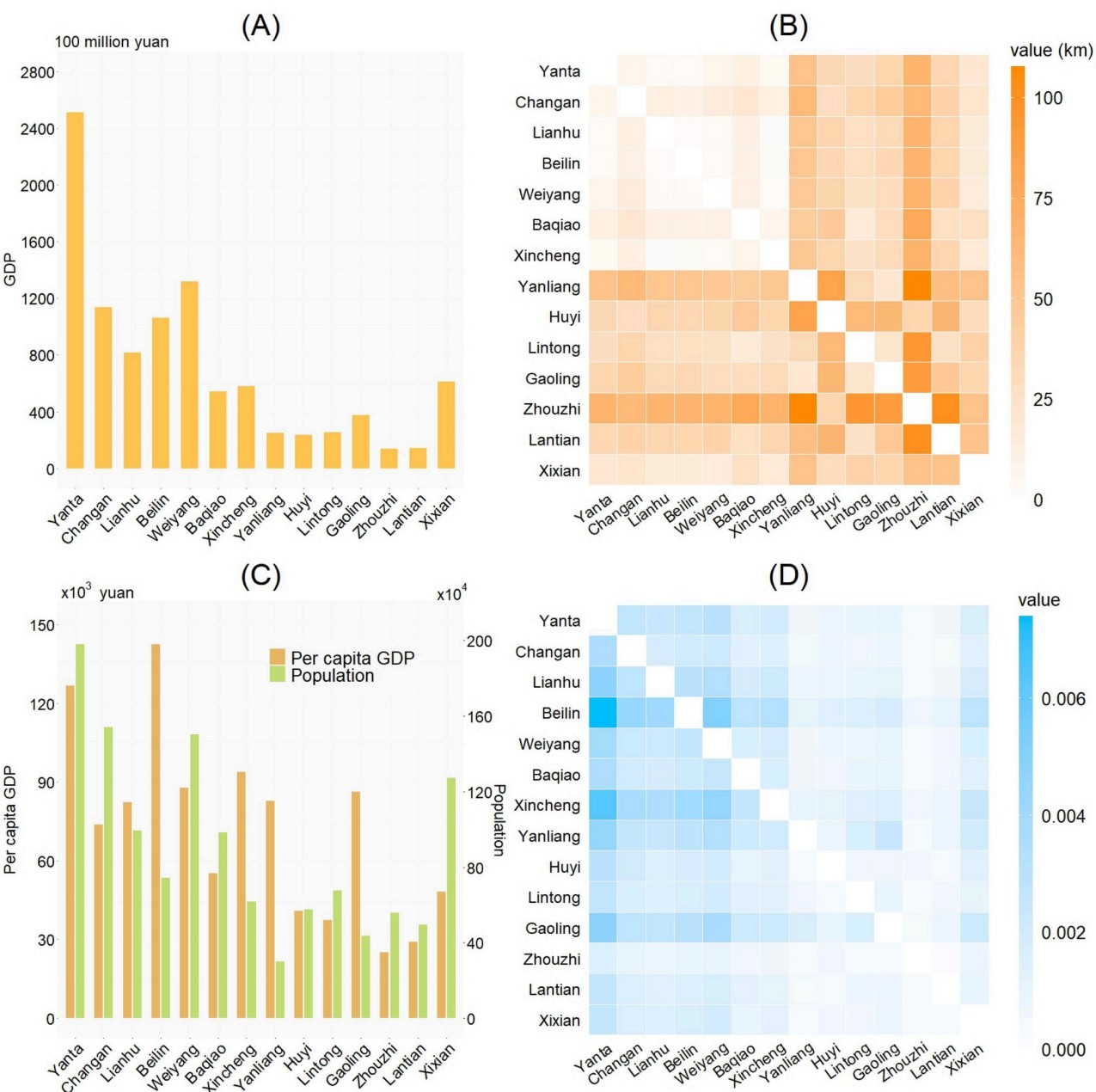

**Fig 4. Demographic, economic and geographic data for districts and counties in Xi'an City.** A: Gross demestic product (GDP) by region (2020). B: Distance between regions. C: Per capita GDP and population by regions (2020). D: Matrix of population migration rate, the entry in the $i$-th row and $k$-th column is the population migration rate $\tau_{ik}$ from region $i$ to region $k$.

individuals in district $i$ are

$$E_i = E_i^m + E_i^s, \quad I_i = I_i^m + I_i^s,$$

respectively, the total number of individuals that can contact others in the district is

$$N_i^f = S_i + E_i + I_i + R_i,$$

and the total population in region $i$ is

$$N_i = N_i^f + C_i^{E,m} + C_i^{E,s} + C_i^{I,m} + C_i^{I,s} + H_i.$$

Let $\sigma$ be the progression rate of exposed individuals to infectives and $\gamma^\omega$ be the recovery rate of mild ($\omega = m$) or severe ($\omega = s$) cases. Let $\beta_i$ be the incidence rate which is the product of the contact rate and the probability of disease transmission in a contact between a susceptible and an infectious subject. The voluntary testing rate, i.e., the diagnostic rate excluding contact tracing and nucleic acid screening, is denoted by $\delta_i$. The subscript $i$ represents the value of the parameter in region $i$.

In this model, we consider the following four non-pharmaceutical interventions (NPIs) that had been implemented in Xi'an during the outbreak: the precision control ($p$, i.e., contact tracing) that puts exposed individuals into quarantine and infectious individuals into isolation; nucleic acid screening ($n$) that detects and isolates infectious individuals; border closure of a district ($b$) that stops population movement between it and other districts; and district lockdown ($l$) that reduces the transmission rate in the district in addition to border closure (i.e., a lockdown implies a border closure). Let $u_i^c(t)$ denote whether a control measure $c = p, n, b, l$ was implemented at time $t$ in district $i$, which takes a value of 1 if it was implemented and 0 if not. We denote the control vector $\boldsymbol{u} = \{u_i^c\}$ for all districit $i = 1, ..v$ and control measures $c = p, n, b, l$.

Assume that the freely moving individuals in compartments $S$, $E$, $I$, and $R$ migrate from district $i$ to $j$ at a rate $\tau_{ij}$ without control measures. Considering border closure and district lockdown, then movement rate from district $i$ to $j$ is

$$m_{ij}(\boldsymbol{u}) = \tau_{ij}(1 - u_i^b)(1 - u_i^l)(1 - u_j^b)(1 - u_j^l),$$

the derivation of the expression of parameter $\tau_{ij}$ is illustrated in section 1 of S1 Appendix. Also assume that the lockdown of a district $i$ at a strength $u_i^l$ reduces the transmission rate $\beta_i$ by a factor $(1 - \varepsilon_i u_i^l)$ where $\varepsilon_i$ is the maximum reduction of the transmission rate with a lockdown.

When nucleic acid screening is executed at some time $t_s$ in a district $i$, a fraction $f$ of the infectious individuals are diagnosed and put into the isolated compartment $C^I$, where $f$ is the false negative rate. This is modeled as a pulse function

$$I_i^\omega(t_s) = I_i^\omega(t_s^-) - u_i^n(t_s)(1 - f) \cdot I_i^\omega(t_s^-), \tag{1a}$$

$$C_i^{I,\omega}(t_s) = C_i^{I,\omega}(t_s^-) + u_i^n(t_s)(1 - f) \cdot I_i^\omega(t_s^-), \tag{1b}$$

for the district index $i = 1, 2, \ldots, 14$ and the severity index $\omega = m, s$. The effectiveness of contact tracing is affected by the outbreak size and decreases when the number of cases exceeds the test-trace-isolate capacity of the local health authority [52–54]. Hence we assume that the contact tracing rate is approximately constant at a maximum rate when the number of cases to be traced is small, and drops sharply when the capacity is reached. This is modeled as a logistically decaying factor $\ell(C^T)$ multiplied to the maximum rate, where $C^T$ is a function of the total number of traced cases. Specifically, let $\lambda_i^E$ be the maximum quarantine rate for the compartment $E$ and $\lambda_i^I$ be the maximum isolation rate for the compartment $I$ in a region $i$. Then

$$L_i^E = u^p \lambda^E \ell(C^T), \quad L_i^I = u^p \lambda^I \ell(C^T), \tag{2}$$

where

$$\ell(C^T) = \frac{1}{1 + \exp(C^T - L^o)}, \tag{3}$$

where $L^o$ is the capacity, specifically, where the $\ell$ drops fastest. At last, the total number of traced cases is proportional to the sum of all cases in each district that implemented contact tracing, i.e.,

$$C^T = r \sum_{i=1}^{v} u_i^p (I_i + E_i).$$

To match model solutions to data, we track the cumulative number of new cases $Y_i(t)$ in district $i$ who were isolated at time $t$. Here new cases come from voluntary testing of individuals in classes $I_i^m$ and $I_i^s$ and contact tracing. Since people who have contacted with a confirmed case will be identified as close contacts and quarantined even though they have not been tested positive, new isolated cases include a certain number of people in latent period.

The flowchart of the model is given in Fig 5, the list of parameters are given in Table 1, and the equations are given in the following system.

$$
\begin{cases}
\dfrac{dS_i}{dt} = \sum_{k \neq i} m_{ki}(\boldsymbol{u})S_k - \sum_{k \neq i} m_{ik}(\boldsymbol{u})S_i - (1 - \varepsilon_i u_i^l)\beta_i \dfrac{S_i I_i}{N_i^f}, \\[2mm]
\dfrac{dE_i^\omega}{dt} = \sum_{k \neq i} m_{ki}(\boldsymbol{u})E_k^\omega - \sum_{k \neq i} m_{ik}(\boldsymbol{u})E_i^\omega + q^\omega(1 - \varepsilon_i u_i^l)\beta_i \dfrac{S_i I_i}{N_i^f} - \sigma E_i^\omega - u_i^p L_i^E E_i^\omega, \\[2mm]
\dfrac{dI_i^\omega}{dt} = \sum_{k \neq i} m_{ki}(\boldsymbol{u})I_k^\omega - \sum_{k \neq i} m_{ik}(\boldsymbol{u})I_i^\omega + \sigma E_i^\omega - (\gamma^\omega + \delta_i)I_i^\omega - u_i^p L_i^I I_i^\omega, \\[2mm]
\dfrac{dC_i^{E,\omega}}{dt} = u_i^p L_i^E E_i^\omega - \sigma C_i^{E,\omega}, \\[2mm]
\dfrac{dC_i^{I,\omega}}{dt} = \delta_i I_i^\omega + u_i^p L_i^I I_i^\omega + \sigma C_i^{E,\omega} - \gamma^\omega C_i^{I,\omega}, \\[2mm]
\dfrac{dH_i}{dt} = \gamma^s I_i^s + \gamma^s C_i^{I,s} - \gamma^H H_i, \\[2mm]
\dfrac{dR_i}{dt} = \sum_{k \neq i} m_{ki}(\boldsymbol{u})R_k - \sum_{k \neq i} m_{ik}(\boldsymbol{u})R_i + \gamma^m I_i^m + \gamma^m C_i^{I,m} + \gamma^H H_i, \\[2mm]
\dfrac{dY_i}{dt} = \sum_{\omega = m,s} (\delta_i I_i^\omega + u_i^p L_i^I I_i^\omega + u_i^p L_i^E E_i^\omega).
\end{cases}
\tag{4}
$$

for $i = 1, 2, \ldots, 14$ and $\omega = m, s$. The initial conditions are $S_i(0) = N_i$, $E_1^m(0) = 1$ (for Yanta District) and $E_i(0) = 0$ for $i \neq 1$, and $I_i^s = I_i^m = C_i^{E,m} = C_i^{E,s} = C_i^{I,m} = C_i^{I,s} = H_i = R_i = 0$ for $i = 1, \ldots 14$.

## Parameters estimation

We fit model (4) to the observed new cases data from December 5, 2021 to January 20, 2022 in districts of Xi'an to estimate the unknown model parameters. The exact dates of the implementation of control measures implemented during the outbreak in Xi'an are published in [47, 48]. This gives use the values of the control vector $\boldsymbol{u}(t)$ for all time $t$, which are shown in the top panels of Fig 3, which red means the corresponding control takes the value 1 (and 0 otherwise). Specifically, contact tracing is implemented on day 5 for all districts, border closure and

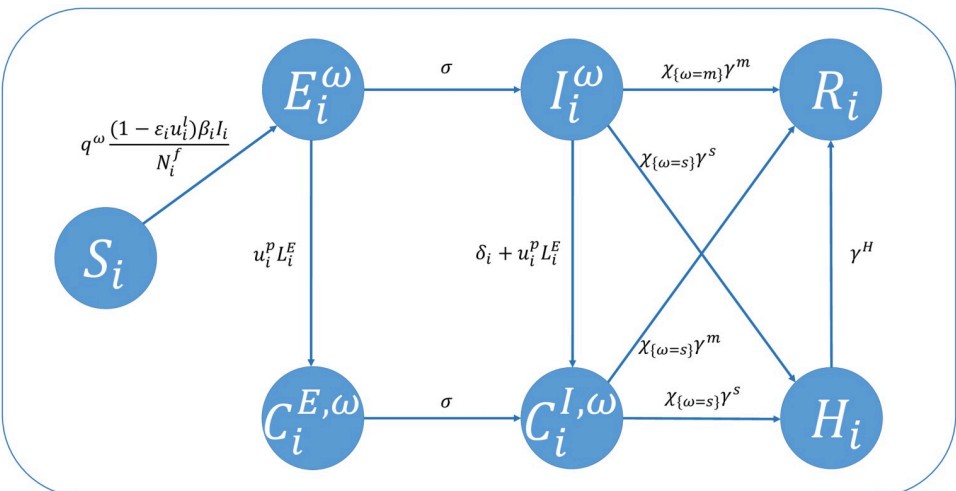

**Fig 5. Diagram of model (4) for some region $i$ in Xi'an City, Shaanxi province, China.** The symbol $\chi_A$ is the index function of set $A$, namely $\chi_A = 1$ when $A$ holds and $\chi_A = 0$ otherwise.

lockdown are implemented on day 18 for all districts, and nucleic acid screening is implemented on day 13 for Yanta district and day 15 for all other districts.

Some parameter values are taken from the literature (see Table 1). The parameters that need to be fitted are denoted as

$$\boldsymbol{\theta} = \{f, \lambda^E, \lambda^I, r, L^o, D, \beta_i, \varepsilon_i, \delta_i, \ \text{for}\ i = 1, 2, \dots, 14\}. \tag{5}$$

These are listed in Table 1 (those labeled as *fitted*). In addition, the migrations rates of the population between districts ($\tau_{ij}$) are estimated using a gravity model [60] and the only

**Table 1. Model parameters and their values in system (4).**

| Parameter | Description | Value (0.95 CI) | Source |
|---|---|---|---|
| $f$ | False negative probability | 0.388 (0.325, 0.449) | fitted |
| $\lambda^E$ | Maximum contact tracing rate of exposed individuals | 0.195 (0.162, 0.227) | fitted |
| $\lambda^I$ | Maximum contact tracing rate of infective individuals | 0.350 (0.296, 0.407) | fitted |
| $r$ | Decreasing rate of the contact tracing efficiency | $2.21\ (1.34, 3.61) \times 10^{-3}$ | fitted |
| $L_o$ | Parameter in contact tracing function (2) | 440 (431, 450) | fitted |
| $D$ | Parameter in commuting flow (see S1 Appendix section 1) | 0.844 (0.646, 1.07) | fitted |
| $\phi$ | Parameter in likelihood function (7) | 1.001 (0.823, 1.192) | fitted |
| $\sigma$ | Progression rate of exposed individuals to infectives | 0.417 | [56] |
| $q^s$ | Probability of patients showing severe symptoms | 1.5% | [57] |
| $1/\gamma^H$ | Hospitalization time of severe cases | 7 days | [58] |
| $1/\gamma^s$ | Time between becoming infectious and hospitalization | 2 days | assumed |
| $\gamma^m$ | Recovery rate of mild cases | 0.432 | estimated |
| $H_{max}$ | Maximum capacity of hospital beds for severe cases | 411 | [59] |
| $\beta_i$ | Transmission rate in district $i$ | See Table 2 | fitted |
| $\varepsilon_i$ | Reduction of $\beta_i$ due to lockdown in district $i$ | See Table 2 | fitted |
| $\delta_i$ | Diagnosis rate in district $i$ | See Table 2 | fitted |
| $\tau_{ij}$ | Migration rate between districts $i$ and $j$ | See Fig 4D | estimated |

**Table 2. The estimated values for regional dependent parameters (95% credible intervals) in model (4), i.e., the transmission rate $\beta_i$, the reduction in the transmission rate due to lockdown $\varepsilon_i$, and the diagnosis rate $\delta_i$.**

| Region | Transmission rate $\beta_i$ | Transmission rate reduction $\varepsilon_i$ | Diagnosis rate $\delta_i$ |
|---|---|---|---|
| Yanta | 2.601 (2.492, 2.706) | 0.689 (0.626, 0.742) | 0.041 (0.022, 0.060) |
| Changan | 2.963 (2.777, 3.176) | 0.684 (0.623, 0.751) | 0.028 (0.008, 0.046) |
| Lianhu | 2.977 (2.795, 3.169) | 0.747 (0.668, 0.821) | 0.029 (0.009, 0.048) |
| Beilin | 2.890 (2.678, 3.096) | 0.772 (0.702, 0.845) | 0.041 (0.021, 0.059) |
| Weiyang | 2.705 (2.495, 2.916) | 0.795 (0.723, 0.870) | 0.032 (0.011, 0.050) |
| Baqiao | 2.469 (2.188, 2.777) | 0.804 (0.717, 0.893) | 0.030 (0.011, 0.049) |
| Xincheng | 2.497 (2.184, 2.782) | 0.746 (0.654, 0.827) | 0.022 (0.003, 0.040) |
| Yanliang | 2.820 (2.515, 3.103) | 0.735 (0.656, 0.818) | 0.029 (0.007, 0.048) |
| Huyi | 2.407 (1.940, 2.873) | 0.774 (0.680, 0.863) | 0.031 (0.011, 0.051) |
| Lintong | 0.821 (0.447, 1.413) | 0.753 (0.663, 0.852) | 0.029 (0.010, 0.048) |
| Gaoling | 0.858 (0.445, 1.540) | 0.756 (0.657, 0.852) | 0.021 (0.004, 0.040) |
| Zhouzhi | 0.797 (0.441, 1.507) | 0.748 (0.659, 0.845) | 0.020 (0.003, 0.038) |
| Lantian | 0.732 (0.449, 1.357) | 0.749 (0.660, 0.851) | 0.021 (0.004, 0.040) |
| Xixian | 2.987 (2.798, 3.188) | 0.694 (0.633, 0.756) | 0.028 (0.010, 0.047) |

unknown parameter in the expression for $\tau_{ij}$ is $D$ in (5), please see the detailed description of the estimation process in section 1 of S1 Appendix.

The new cases in district $i$ on day $t$ is thus

$$\mu_i(t; \boldsymbol{\theta}) = Y_i(t) - Y_i(t-1).\qquad(6)$$

Note that $Y_i(t)$ implicitly depends on parameter values $\boldsymbol{\theta}$.

We adopt the commonly used assumptions in literature that the number of observed cases $x_i(t)$ in district $i$ on day $t$ is independently negative binomial distributed with a mean $\mu_i(t)$, where the size parameter $\phi$ is fitted together with other model parameters in $\boldsymbol{\theta}$ [4, 20, 56, 57]. This distribution is chosen because that over-dispersion often occurs in epidemiological count data [57, 58]. Thus, the likelihood function for observing the data $\{x_i(t)\}_{i=1}^{14}$ given the parameters $\boldsymbol{\theta}$ (which now includes the size parameter $\phi$) is

$$P_{nb}(\{x_i(t)\}_{i,t} | \{\mu_i(t)\}_{i,t}, \phi) = \prod_{i,t} \frac{(x_i(t) + \phi - 1)!}{x_i(t)!(\phi - 1)!} \left(\frac{\mu_i(t)}{\mu_i(t) + \phi}\right)^{x_i(t)} \left(\frac{\phi}{\mu_i(t) + \phi}\right)^{\phi},\qquad(7)$$

where the date $t$ ranges from December 5, 2021 ($t = 0$) to January 20, 2022 ($t = 46$). By Bayes' theorem [64] the posterior distribution of $\boldsymbol{\theta}$ is proportional to the product of the prior distribution and the likelihood function. The construction of the prior distribution is shown in section 2 of S1 Appendix. We obtain parameter estimates from the posterior sample obtained by sampling from the posterior distribution using Markov Chain Monte Carlo (MCMC) method [66].

We conducted MCMC sampling of the posterior distribution using the adaptMCMC package in the R programming language (version 4.1) [66]. We ran MCMC for $2 \times 10^5$ iterations with $1 \times 10^5$ iterations of warm-up and a thinning factor 100 to obtain 1000 posterior samples of parameter vector $\boldsymbol{\theta}$. We take the posterior median as the point estimate, which is referred to as the baseline model parameters to be used to identify optimal control measures.

## Optimal control measure

The main goal of this paper is to develop a framework to identify optimal control strategies that carry a wide range of socio-economic costs. The optimal strategy is the one with the minimum cost, provided that disease control is eventually achieved. Thus, we setup an optimal control problem where the objective function is the total socio-economic cost from the detection of the first case to the end of the outbreak, and the constraints are that the cases eventually approaches 0, and number hospitalized patients in any day does not exceed the hospital capacity. The control variables are $\boldsymbol{u}(t)$, i.e., whether each control measure is implemented at time $t$.

We first list the constraints. The primary goal for the interventions is to control the outbreak by a given time $t_f$, i.e.,

$$\sum_{i=1}^{14} \sum_{\omega=m,s} (E_i^{\omega}(t_f) + I_i^{\omega}(t_f)) = 0.$$

For Xi'an City, the outbreak started on December 05, 2021 ($t = 0$), and ended on January 20, 2022 ($t = 46$). the optimal control strategy cannot be worse than the strategy implemented during the outbreak, when was controlled in 46 days. Thus, $t_f = 46$. In addition the number of hospitalized patients at any time must not exceed a capacity $H_{max}$, i.e., for all $0 \leq t \leq t_f$,

$$\sum_{i=1}^{14} H_i(t) \leq H_{max}.$$

The value of $H_{max}$ is shown in Table 1.

To establish the objective function $J(\boldsymbol{u})$, we first quantify the costs of each control measure. We assumed that the cost of implementing a control measure per unit time in an area is proportional to the population size. Let $cost_i^c$ be the cost of implementing a control measure $c$ in region $i$ for the duration of the outbreak, then

$$cost_i^c(\boldsymbol{u}^c(t)) = \alpha_i^c \int_0^{t_f} N_i(t)u_i^c(t)dt, \quad c = l, b, n, \tag{8}$$

where $\alpha_i^c$ is the per capita cost of implementing control measure $c$ per unit time in region $i$. The unit of the costs is Chinese Yuan. Since nucleic acid screening is impulsive transformation (1) at times $t_s$, for $s = 1, 2, \ldots$, the cost for it is simplified as

$$cost_i^n(\boldsymbol{u}^n(t)) = \alpha_i^n \sum_{s=1,2,\ldots} N_i(t_s). \tag{9}$$

where the implementation time $t_s$ are when $u_i^n(t_s) = 1$.

Since the scale of contact tracing is much smaller than that of lockdown, regional border closure, and nucleic acid screening, it costs much less than the other control measures. Therefore, we do not include the cost of contact tracing in total cost for simplification and assume that contact tracing is always present once the outbreak is detected ($u_p \equiv 1$). Thus, we set the total cost of control measures during the outbreak as

$$J(\boldsymbol{u}(t)) := \sum_{i=1}^{14} \sum_{c=l,b,n} cost_i^c(\boldsymbol{u}^c(t)). \tag{10}$$

During a lockdown, most residents except those in crucial services stop working and stay home, therefore we assume that the cost of lockdown is the lost labor production value and

take the cost coefficient $\alpha_i^l$ to be the daily per capita GDP in region $i$ (shown in Fig 4C). The cost of nucleic acid screening on each individual is assumed to be the charge per person per nucleic acid test during the epidemic, and the cost coefficient is chosen to be $\alpha_i^n = 10$ according to the announcement by the National Healthcare Security Administration of China [65]. Since there is no data for the coefficient of regional border closure cost $\alpha_i^b$, we take $\alpha_i^b = 5\%\alpha_i^l$, and then perform a sensitivity analysis on the value.

The constrains include that the number of hospitalized patients in the city at any time does not exceeds a capacity $H_{max}$. Based on transmission and control model (4), (1) and control cost function (10), we formulate the following OCP:

$$\min_{\mathbf{u^l},\mathbf{u^b},\mathbf{u^n}} \quad J(\mathbf{u}(t)) := \sum_{i=1}^{14}\sum_{c=l,b,n} cost_i^c(\mathbf{u}^c(t)),$$

$$\text{s.t.} \quad \sum_{i=1}^{14} H_i(t) \le H_{max}, \ \ t \in [0, t_f], \tag{11}$$

$$\sum_{i=1}^{14}\sum_{\omega=m,s}(E_i^\omega(t_f) + I_i^\omega(t_f)) = 0,$$

where $u_i^c(t) = 0$ or 1, and $H_i$, $E_i^\omega$ and $I_i^\omega$ are solutions of (4). Note that in the solving of OCP (11), for control measures $c = p, n, b, l$, we determined the earliest time $t_i^c$ for implementing $c$ in region $i$ based on the different stages of disease transmission and epidemic control in reality. Before the time $t_i^c$, control variable $u_i^c(t)$ can only be 0. Moreover, starting from a positive initial condition, the state variables of the epidemic model (4) can only converge to zero asymptotically, rather than become zero in finite time. Therefore, in order to make the last constraint in OCP (11) theoretically achievable, we established a mapping to determine whether $E$ and $I$ can be regarded as zero when their values is very small. See section 4 of S1 Appendix for details.

System (11) was solved using a simulated annealing algorithm [67] conducting in the R programming language (version 4.1), the details and the convergence of the algorithm are illustrated in section 9 of S1 Appendix.

## Results

### Parameter estimation

Before using the method described in section Parameters estimation to estimate model parameters, we first estimated the recovery rate $\gamma$ by fitting the distribution of the model-derived generation time to the distribution of the generation time from the literature [59], and the estimated value of $\gamma$ is shown in Table 1. For details of the estimation, please see section 3 of S1 Appendix.

The estimates of the remaining parameters were obtained by the Bayesian model and MCMC method in section Parameters estimation and are shown in Table 1. All model parameters are identifiable. Fig 2 shows that the estimated parameter values yield a model solution that agrees well with observed cases. S1 Fig shows the same comparison with different vertical scales to show data more clearly. The estimated population migration rate matrix $(\tau_{ik})$ is shown in Fig 4D.

## The effect of control timeliness and disease transmissibility on the epidemic

To investigate the effect of timely implementation of control measures and the effect of transmissibility of the disease on the epidemic, we simulated the epidemic under different scenarios by varying the starting time of control measures and values of parameters related to disease transmissibility.

To study the effect of the time of implementation of control measures on outbreak, in this section we assume that the implementation of control measure $c$ can be advanced to the end of hidden transmission stage $t_i^p$ at the earliest. It is assumed that contact tracing is always implemented on the day of index case detection (i.e. the time $t_i^p$). We vary the dates of implementation for the other control measures from the actual by $d_i^c$ days for nucleic acid screening ($c = n$), border closure ($c = b$) and lockdown ($c = l$) for district $i$. Here $d_i^c > 0$ means a delay of $d_i^c$ days, and $d_i^c < 0$ means an earlier implementation.

In addition to the baseline control scheme $U_0 = \{d_i^n = d_i^b = d_i^l = 0\}$, we also consider 4 addition control schemes: each of the three control measures $n$, $b$, $l$ is shifted by $d$ days individuals while holding the others unchanged, namely, $U_1 = \{d_i^n = d, d_i^b = d_i^l = 0\}$, $U_2 = \{d_i^b = d, d_i^n = d_i^l = 0\}$, $U_3 = \{d_i^l = d, d_i^n = d_i^b = 0\}$, and all three control measures are shifted by $d$ days, namely, and $U_4 = \{d_i^n = d_i^b = d_i^l = d\}$. Results are presented in Fig 6.

Fig 6A1 to 6A3 show the prevalence at time $t$, namely $I(t) := \sum_{i=1}^{14}(I_i^m(t) + I_i^s(t))$, corresponding to control schemes $U_1$, $U_2$ and $U_3$, respectively. Another measure of the severity of an epidemic is the final epidemic size. In a district $i$, this is defined as

$$Cum_i := \int_0^{t_f}(1 - \epsilon_i u_i^l)\beta_i \frac{S_i I_i}{N_i^f}\,dt,$$

while the final epidemic size for the city is

$$Cum := \sum_{i=1}^{14} Cum_i.$$

Fig 6A4 shows the epidemic size for the city as a function of $h$ for $U_1$–$U_4$. Note that the prevalence $I(t)$ is a discontinuous function of time $t$ because the nucleic acid screening are implemented in pulses, which removes a fraction of $I(t)$ in each pulse. It can be seen from Fig 6A1 and 6A4 that the number of infections decreases with earlier nucleic acid screening. Similarly, Fig 6A3 and 6A4 shows a similar effect for lockdown, and starting lockdown one day earlier is more effective in controlling the outbreak than starting nucleic acid screening one day earlier. It is worth noting that implementing regional border closure a few days earlier does not impact the epidemic significantly, but an early enough regional border closure causes a significant decrease in the number of infections as shown in Fig 6A2. In section 6 of S1 Appendix, we show $I_i(t)$ for different regions $i = 1, 2, \ldots, 14$ and different values of $d^b$, from which we can see that the phenomenon is due to the fact that early enough implementation of regional border closure prevents the importation of cases from Yanta District to other districts. Note that although earlier implementation of regional border closure does not reduce the number of infections as much as earlier lockdown or nucleic acid screening as shown in Fig 6A4, it prevents the importation of infections into some regions and therefore avoids the cost of implementing control in these regions.

Fig 6B1 to 6B4 shows the effect of changing the transmissibility on the prevalence $I(t)$ and cumulative infections $Cum$ in the city. The transmissibility parameters we consider include the basic reproduction number $\mathcal{R}_0 := \frac{\beta}{\gamma}$ representing the average number of secondary infections

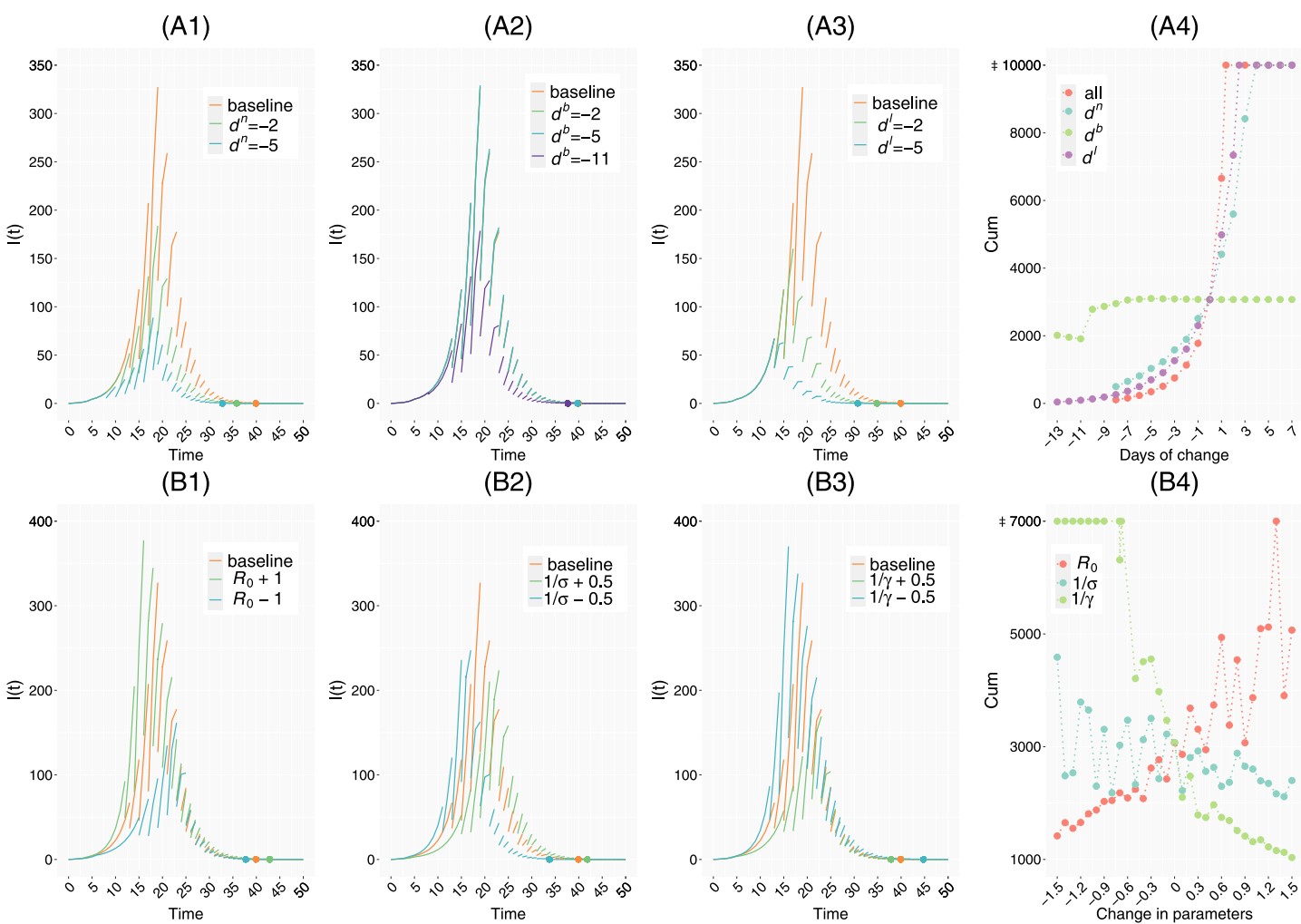

**Fig 6. Simulation results.** A1-A3: Number of infected individuals in the infectious period in all regions $I(t)$ under different control schemes. 'baseline' represents baseline control scheme $U_0$, $d^c$ is the change in the start time of the control measure $c$ relative to $U_0$, with negative values representing advancement. The dots on the horizontal axis represent the time when $I(t)$ becomes 0. A4: Cumulative infections $Cum$ under different control schemes. The horizontal coordinate represents days of change $d^c$ in start time of control $c = n, b, l$. 'all' indicates that all three control measures $n, b, l$ are varied simultaneously, and '$d^c$' indicates that only control measure $c$ is varied. B1-B3: $I(t)$ under different parameters of transmissibilty. 'baseline' denotes baseline value of parameters, '$R_0 + a$' means to change $\beta_i$ to $a\gamma + \beta_i$ for all $i$ (thus changing $R_{0,i} := \frac{\beta_i}{\gamma}$ to $R_{0,i} + a$), '$\frac{1}{\sigma} + a$', '$\frac{1}{\gamma} + a$' means to adjust $\sigma, \gamma$ to increase the original latent period $\frac{1}{\sigma}$ and infectious period $\frac{1}{\gamma}$ by $a$ days. B4: Cumulative infections with respect to the change $a$ in parameters of transmissibilty, with the horizontal coordinate representing the change $a$. Break points in $I(t)$ curves are due to the impulsive effect (1).

produced by an infectious individual in a fully susceptible population, the mean latent period $\frac{1}{\sigma}$ and mean infectious period $\frac{1}{\gamma}$. Note that when we vary the value of $\gamma$, we also change the value of $\beta$ accordingly to keep the basic reproduction number $\mathcal{R}_0$ constant. From Fig 6 we can see that the increase of the basic reproduction number $\mathcal{R}_0$, the shortening of the latent period or the shortening of the infectious period lead to an increase in the number of infections, and the infectious period has a greater effect than the latent period for the same change in the number of days. It should be noted that the time taken for the outbreak to grow large enough to be detected varies with these parameters, therefore we did not fix the starting time of control measures in the simulation presented in Fig 6B1 to 6B4, instead we determined the starting time according to the number of infections or confirmed cases and the details are illustrated in section 7 of S1 Appendix. In addition, since we assume that control measures are implemented at

integer time points, three curves in Fig 6B4 are not continuous. What we really need to focus on is the overall trend of these curves, rather than their fluctuation.

## Population mobility across regions and risk of epidemics importation

From Fig 4D we can see that Yanta, Chang'an, Lianhu, Beilin and Weiyang have higher population inflow rates, followed by Baqiao District, Xincheng District and Xixian New Area, while the other regions have lower population inflow rates. With the population migration rate and the resident population of Yanta District shown in Fig 4, we can get the daily population input $N_1\tau_{1,k}$ from Yanta District to any other region $k$, as shown in Fig 7A.

Model (4) is a deterministic system. However, the real disease outbreak is a integer-valued stochastic process. A stochastic models can more realistically characterize the disease transmission when the number of infected individuals is small [65]. Considering stochasticity is more important in during the importation phase, when the case counts were low. Therefore, in order to further study the effect of stochasticity in case importation, we developed a continuous time Markov chains (CTMC) model corresponding to the ODE model (4) to simulate the epidemic. For details about CTMC modelling and numerical simulations, please refer to section 8 of S1 Appendix.

We simulated the outbreak in the absence of large-scale control measures to obtain the importation time for each region shown in Fig 7B, and simulated the outbreak with the baseline control scheme $U_0$ and with control scheme $U(0, d^b, 0)$ which advances border closure in all regions by $d^b$ days on the basis of $U_0$ to obtain Fig 7C to 7D.

Fig 7B shows the median importation time of the first imported case in each region, which agrees with our intuition that the larger $N_1\tau_{1,k}$ the earlier the importation of infections into region $k$. We show the probability that a region has imported infected individuals in Fig 7C, from which we can see that in the absence of control measures, the probability of infection importation is almost the same for each region, which is about 70%, and when there are control measures in place, the probability of infection importation increases with $N_1\tau_{1,\cdot}$, i.e., the greater the population inflow from Yanta to a region, the greater the probability of infection importation of this region. It can also be seen that the earlier the border closure is implemented, the greater the probability of no case importation in each region. When the border closure is implemented more than 10 days earlier than the baseline, almost all regions can avoid the importation of the outbreak with a higher probability, which can explain the results in Fig 6A2, i.e., timely border closure can avoid epidemic outflow from Yanta District, and thus play a significant role in controlling the epidemic. Fig 7D shows the mean number of imported cases into each region, from which we can see that for any region $k$, the larger $N_1\tau_{1,k}$ the more the imported cases into region $k$, and the earlier the control time the less the imported cases.

## Optimal control for the epidemic and influencing factors of control cost

The optimal solution for control variables $\{u_i^c\}$ is shown in Fig 3 bottom panel, where the rows represent regions, the columns represent time; a red block means that the control measure is implemented in the region on that day. The total cost of the optimal control strategy for the city is 33.4 billion yuan, note that this is significantly less than the estimated cost of the control measures actually implemented during the outbreak (85.7 billion yuan).

From Fig 3 we can see that if the requirement is to clear the outbreak within a short period of time using NPIs, control measures should be implemented as early as possible and continuously in any region that has cases until the outbreak ends. When the scale of outbreak is large, significant control measures such as lockdown and nucleic acid screening should be

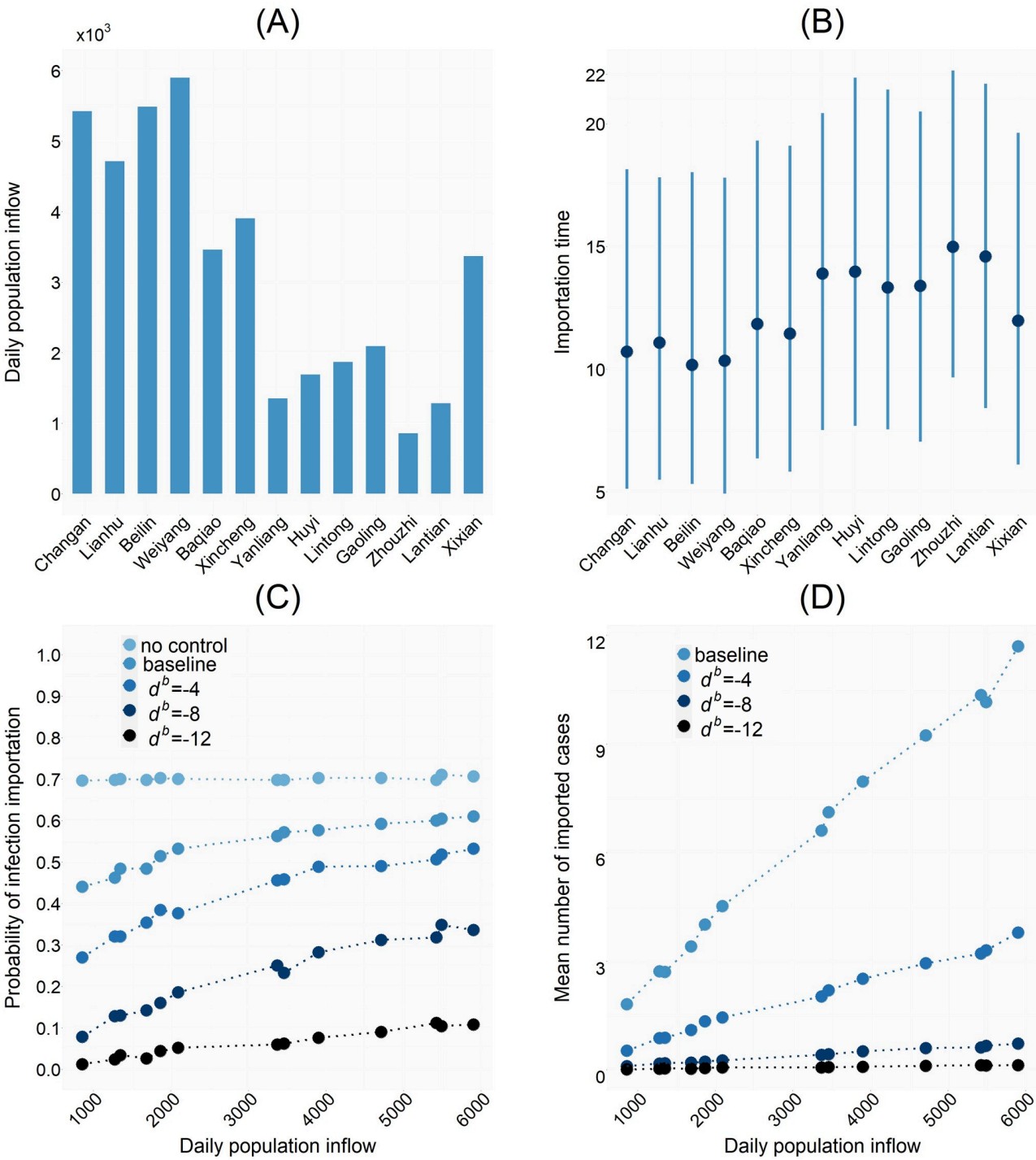

**Fig 7. Simulation results.** A: Daily population inflow from Yanta District to other regions. B: Median, 10% quantile and 90% quantile of the earliest time of emergence of infections in each region in the absence of large-scale control measures. C-D: Variation of the infection importation probability and the number of imported cases about daily population inflow from Yanta District under different control schemes.

implemented simultaneously to control the outbreak. Compared to control measures actually implemented in Xi'an City as shown in Fig 3 top panel, the optimal control strategy implements universal nucleic acid screening more frequently in the early phase of the epidemic. This implies that the combination of lockdown and nucleic acid screening can control the epidemic more effectively and with lower cost. Fig 3 also shows that the optimal control has more significant regional heterogeneity, i.e., the duration of control varies significantly across districts. In particular, control measures of lockdown, district border closure and universal nucleic acid screening are not necessary for Lintong, Gaoling, Zhouzhi and Lantian during the outbreak. This suggests that the cost of control can be greatly reduced by implementing control measures according to the severity of the epidemic and the population and economic conditions of each region.

According to our assumption, the economic loss caused by lockdown in a region is related to the local GDP, and there are significant differences in the magnitude of GDP among different regions. Therefore, if we directly compare the amount of control costs in different regions, it is more likely to observe significantly higher control costs in economically developed regions compared to economically underdeveloped regions, even if the former implement control measures for a shorter duration. Consequently, in order to compare the cost of controls across different regions under the same magnitude, we defined the relative control cost for the optimal control and the actual control of each region $i$ as the amount of control cost divided by the regional annual GDP:

$$p_i^{op} = \frac{cost_i^{op}}{\text{GDP}_i} \quad \text{and} \quad p_i^{ac} = \frac{cost_i^{ac}}{\text{GDP}_i}, \tag{12}$$

here $cost_i^{op}$ and $cost_i^{ac}$ are the cost of the optimal control scheme and the actual control scheme respectively. Values of $cost_i^{op}$, $cost_i^{ac}$, $p_i^{op}$, $p_i^{op}$ are shown in Fig 8A, from which we can see that the relative costs of actual control scheme $p_i^{ac}$ are similar for all regions (about 8.6%), while the relative costs of optimal control scheme $p_i^{op}$ are significantly lower than them. The relative cost of optimal control is the highest in Yanta District, which is 5.06%. Among the other regions, Changan District, Lianhu District, Beilin District, Weiyang District, and Xixian New Area have higher relative optimal cost that ranges from 2.5% ∼ 4.0%. Lintong District, Gaoling District, Zhouzhi County, and Lantian County have the lowest optimal cost which is 0%, i.e., our model predicts that control measures do not need to be in these regions. The relative cost of optimal control of the remaining regions are about 2%. From Fig 8A, it can be seen that all regions can reduce the control cost by more than 40%, and the smaller the outbreak size, the greater the percentage reduction in the control cost, which indicates that designing the control scheme according to the characteristics of a region can greatly reduce the cost of control.

Intuitively, a district with a larger $\mathcal{R}_0$ or a larger migration rate is harder to control. This is demonstrated by Fig 8B, which shows the relationship between the basic reproduction number $\mathcal{R}_0$, the GDP (affecting the migration rate) and the relative optimal control cost $p_i^{op}$ for each district. It can be seen that $p_i^{op}$, $\mathcal{R}_0$, GDP have similar regional variability; specifically, districts with larger $\mathcal{R}_0$ and GDP have larger optimal control cost. Although $\mathcal{R}_0$ directly affects the epidemic spread, GDP data are more easily accessible. Thus, we may use GDP as a proxy to assess variations of the cost of controlling outbreaks and the size of outbreaks. To further illustrate this, we fit a linear relationship between the relative optimal control cost $p_i^{op}$ and the GDP, and a linear relationship between the final size $Cum_i$ and the GDP. Note that, Lintong, Gaoling, Zhouzhi, Lantian has control cost of zero because either there is no case importation or the epidemic in these regions can be cleared by precise control only. Yanta district as the origin of the outbreak has its peculiarities, so we did not include these 5 regions in the regression model

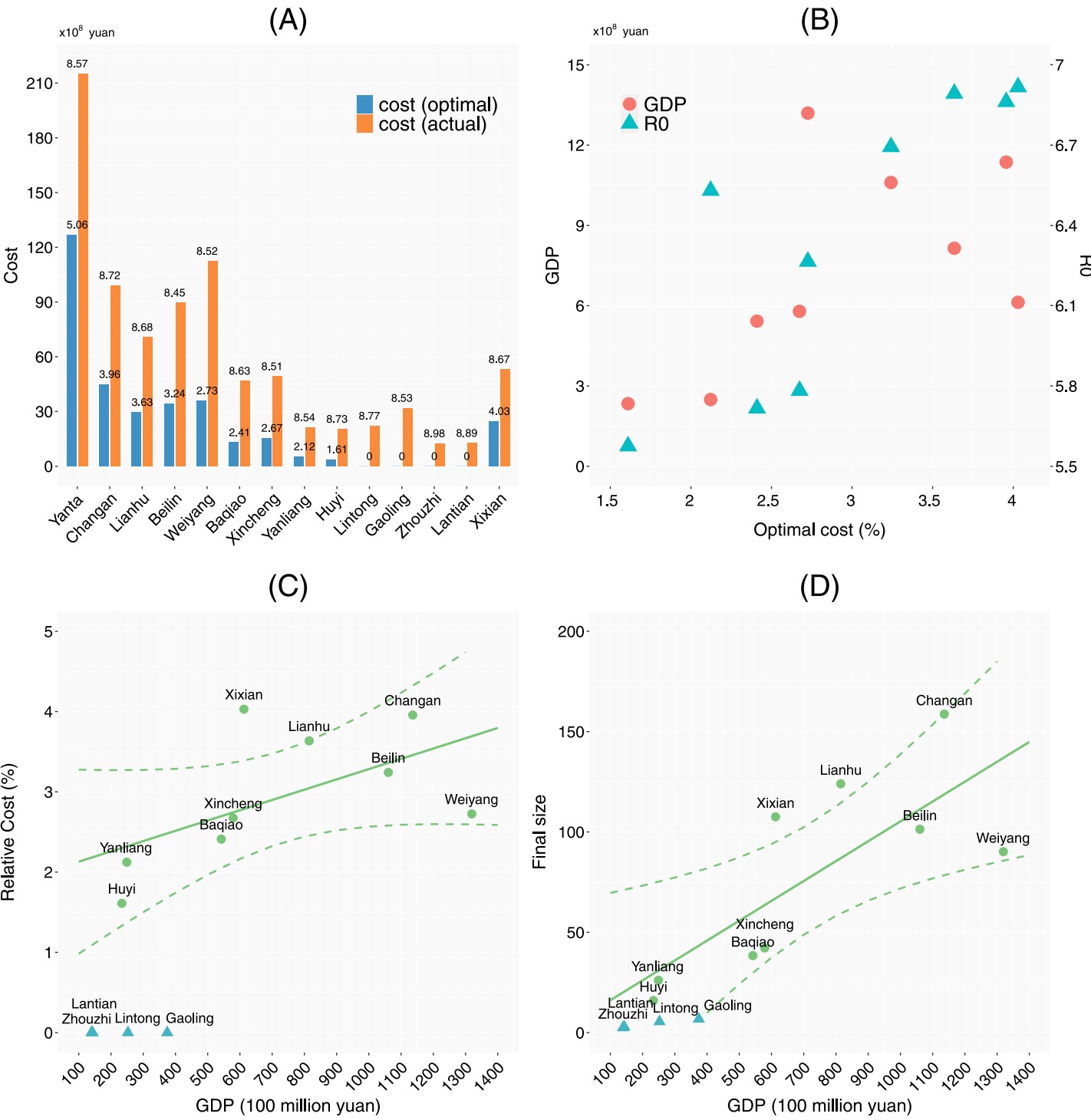

**Fig 8. Control costs and their influencing factors.** A: The cost of the optimal control scheme (red bars) and the cost of the actual control scheme (blue bars). The number above a bar is the relative control cost (i.e., proportion of control cost to the regional annual GDP (percentage)). B: Relationship between the relative optimal control cost, the basic reproduction number $R_0$, and GDP in regions except Yanta, Lintong, Gaoling, Zhouzhi, Lantian. C-D: Relationship between the relative optimal control cost, the final size and GDP. The solid line represents the fitted value of the linear regression, and the dashed line represents the confidence interval of the fitted value. Green points were used in the fitting of regression models and blue points were not involved in the fitting.

fitting. The results of the linear regressions are shown in Fig 8C to 8D. They show that for every 10 billion yuan rise in regional GDP, the cost of optimal control increases by 0.13% and the final size of the outbreak increases by 10 individuals.

Sensitivity analyses with $\alpha_i^b = 2\%\alpha_i^l, 10\%\alpha_i^l, 20\%\alpha_i^l$ indicate that the optimal control scheme of problem (11) is insensitive to these values of $\alpha_i^b$, therefore only the result of $\alpha_i^b = 5\%\alpha_i^l$ is shown here.

## Formulation of the control scheme in the outbreak

After the disease is introduced into a city, the epidemic first enters the stage of hidden transmission, during which the disease spreads freely without any control. When the first confirmed case is identified by the regular epidemic monitoring mechanism, the epidemic will enter the stage of precise control, during which precise control measures such as contact tracking will be carried out. Depending on the epidemiological surveillance data obtained in the precise control, regional governments determine whether to implement large-scale control measures. If it is determined that precise control cannot control the spread of the epidemic, then the epidemic enters the stage of large-scale control, during which lockdown, border closure and massive nucleic acid screening are implemented. After the large-scale controls end, the epidemic enters the stage of regular epidemic control, in which all control measures are lifted except the regular epidemic monitoring mechanisms. The key point of regular prevention and control is the early detection of future imported cases after the epidemic is over. And the key point of precise control is to accurately and quickly identify close contacts of confirmed without introducing large-scale control to reduce the cost of disease control. The specific mathematical description of regular control and precise control are not studied in depth in this paper, which are independent issues worthy of further study.

The bottom panel of Fig 3 shows that the optimal implementation time of nucleic acid screening in Beilin district flips on and off from December 28, 2021 to January 2, 2022. This phenomenon can only be seen in control measures with relatively low cost, and such discontinuous control schemes is not practical. Thus, we suggest a continuous implementation of nucleic acid screening in this period. The cost of resulting control schemes is 33.8 billion yuan, which is only one percent higher than the original optimal control cost.

It should be noted that the control measures in the optimal strategy stops when the number of infected individuals in the isolated regions reached zero (as shown in S2 Fig). However, in reality, the infected individuals cannot be observed directly. They can only be indirectly indicated by the number of confirmed cases. Due to the fact that nucleic acid screening may be false negative, not all infected individuals can be identified by testing. Thus the time when the number of confirmed cases first become zero is often earlier than the time when the number of infected people become zero. To ensure that there is no infected individuals out of isolation area, large-scale control measures should continue to be implemented in a period after the number of confirmed cases reaches zero. In this period, lockdown can be released as early as possible because of its higher cost. That is, assuming the latest time that the number of confirmed cases is greater than zero is $T_i^0$, the lift time of lockdown is set as

$$T_i^l = T_i^0 + T^1, \tag{13}$$

and the lift time of border closure and massive nucleic acid screening is set as

$$T_i^b = T_i^n = T_i^0 + T^2, \tag{14}$$

where $T^1$, $T^2$ are positive integers that need to be determined, and $T^2 \geq T^1$. As shown in S2 Fig, in our simulation, the time when the number of new confirmed cases becomes zero in

each region is at most 7 days earlier than the time when the number of infected individuals becomes zero, hence we can set $T^1 = T^2 = 7$. This is consistent with the Chinese guideline for the prevention and control of COVID-19, which suggests that high-risk regions could be downgraded to medium-risk areas and lockdown can be lifted in these regions if there are no new confirmed cases in 7 consecutive days, and if there are no new confirmed cases in the following 3 consecutive days, these regions can be downgraded to low-risk regions and regional border closure, massive nucleic acid screening can be lifted [69], which indicates $T^1 = 7$ and $T^2 = 10$. An important cause for false negative nucleic acid tests is the low amounts of virus concentrations in patients while incubating, and the viral load and viral shedding always peaks around the onset of symptoms [70]. Therefore taking $T^1 = 7$ and $T^2 = 10$ is reasonable and consistent with the fact that incubation period of COVID-19 less than seven days [71]. That is, in the actual prevention and control, $T^1$ and $T^2$ should be longer than the incubation period to ensure that there is no infected individual out of isolation area when the lockdown is lifted. Since the implementation cost of regional border closure and nucleic acid screening is relatively lower than that of lockdown, $T^2$ can be appropriately longer than $T^1$.

## Conclusion and discussion

There have been many infectious diseases that seriously endanger human health in history, such as Bubonic plague, the 1918 influenza, SARS, and COVID-19. This paper takes COVID-19 as an example to study issues of the control of emerging infectious diseases. Although the policy on COVID-19 has shifted to regular prevention and control, there is still a possibility of serious outbreaks of other diseases in the future. The results in this paper are also applicable to general emerging infectious diseases and therefore provides a reference for responding to possible epidemics.

Before pharmaceutical interventions were widely available, NPIs were crucial to control the COVID-19 pandemic to a manageable level. The most effective NPIs, such as lockdown and border closure, will inevitably incur a high cost on the local economy and people's livelihood, so it is important to reduce the economic loss and negative impacts of control while effectively controlling the epidemic. Thus when and how to implement these control measures relies on a fine balance of between the public health threat and citizen life. The modeling framework presented in this paper provides such a tool to identify the optimal strategy that minimizes economic impact while achieving the goal of disease control. Our framework realistically considers population movements across regions, and regional heterogeneity in epidemic and social factors, allowing regionally heterogeneous implementations of control measures.

Using the COVID-19 outbreak in Xi'an city between December 2021 and January 2022 China as an example, we calibrated the model to the daily cases and hospitalization data, incorporating regional characteristics such as diagnosis rate and contact tracing rate. The estimated regional independent parameter values are consistent with the relevant literature. For example, The value of the basic reproduction number $R_0$ of delta variant in existing studies is mostly in the range of 5 to 7 [72–74], and our estimates of $R_0$ in most regions in Xi'an City is in the range of 5.5 to 7. The false negative rate of nucleic acid test is estimated to be 0.388 in our study and is in the range of 0 to 0.5 in similar studies [75]. The proportion reduction in contact rate $\epsilon_i$ after lockdown is estimated to be 0.69 to 0.8, which is consistent with the existing researches of 0.7 to 0.8 [76, 77]. Since the different demographic and economic conditions of regions may lead to difference in extent of implementing control measures and compliance of residents, we assumed that the parameter $\epsilon_i$ which represents the impact of lockdown can take different values across regions and estimated $\epsilon_i$ by fitting the epidemic data. According to the

results of estimation (Table 1), although the percentages of contact rate reduction are not the same in all regions, they are all around 70%, which is not a particularly large difference.

By solving the optimal control problem, we found that if precise control measures such as contact tracing are not sufficient to effectively control the outbreak, the least costly way to control the outbreak is to implement the most stringent control measures continuously until the epidemic is cleared. The optimal control scheme obtained here requires significantly lower cost than the actual control cost for the epidemic in Xi'an City. It is interesting to note that the optimal control scheme we obtained has significant regional differences, with more permanent control in regions where the epidemic was imported earlier or spread more rapidly, and more short-term control in the other regions. The regional heterogeneity of optimal control scheme is strongly correlated with regional differences in population and economic conditions. Specifically, regions with higher GDP have larger scale of epidemic and higher control cost. The reason for this phenomenon may be that regions with higher GDP have higher urbanization levels and denser populations, and therefore higher population contact rates, resulting in a greater basic reproduction number of infectious disease. And the more developed regions have more frequent population movement with other regions, which also leads to easier importation of epidemic. Therefore, the more populated and economically developed regions have a higher risk of epidemic importation and outbreak. It is more difficult to control the outbreak after it emerges in these regions, and it will take longer time and higher cost to end the epidemic. These regions need a better regular surveillance mechanism to ensure timely detection of imported infections and more medical resources to cope with the larger scale of infection after an outbreak.

We further discussed the effect of timely implementation of control measures and model parameters associated with the transmissibility of the disease on epidmic control. The results indicate that implementing control measures earlier can end the outbreak in a shorter time and with a lower number of cumulative infections. For the transmissibility parameters, an increase in the basic reproduction number $R_0$ or a decrease of the latent period or the infectious period leads to a delay in the ending time of the epidemic and an increase in the number of cumulative infections. The basic reproduction number represents how many second-generation infections an infected person can make, so its effect on the epidemic is explicit, while the shorter latent period or infectious period results in a faster rate of disease transmission relative to the rate of detection of infected individuals, which results in more transmissions per unit time before being diagnosed and isolated. The basic reproduction number, the latent period and the infectious period are important indicators of the ability of a virus to spread in a population, and the above results suggest that diseases with greater transmissibility are more difficult and costly to control. This may also explain the differences in the effectiveness of control against outbreaks caused by different variants in some regions. For example, Shanghai City, China has been known for its precise control of multiple imported epidemics of COVID-19 between 2020 and 2021, but had to implement a two-month lockdown to control the epidemic in early 2022 when encountered the outbreak dominated by the COVID-19 Omicron variant [78].

We adopted a commonly used gravity model [21, 50] that considers GDP and population sizes to describe population flow across regions. This approach reduces the number of parameters to be calibrated. Other approaches to infer population movement include traffic data [22–24] and mobile phone data [25]. However such data are not available for most researchers. Other factors such as climatic and geographic conditions can also lead to regional differences in disease transmission [79, 80], which are only implicitly considered in this model using regionally heterogeneous transmission rate. Considering these factors explicitly significantly

increases model complexity, risking to reduce parameter identifiability. The fact that our model fitted the data very well demonstrates that this is a parsimonious yet effective approach.

Even though this study uses a specific city as an example the model is generalizable to other regions. The cost function for the optimal control was constructed relatively preliminarily and might not completely equal to the economic loss caused by epidemic control. In fact, the economic cost caused by the epidemic is an economic issue worth studying [81–84]. Due to a different focus in this paper, we did not discuss in depth how to quantify the economic loss during the epidemic more precisely, but the main conclusions of this paper remain the same if the relative orders of magnitudes of costs incurred by different control measures are similar in actuality and our cost function (10). In addition, this modeling framework can be easily extended to consider other socio-economical and political costs such as excess deaths, population tolerance and policy adherence.

In this modeling framework, eventually curtailing the outbreak is a condition in the optimal control problem. It can be easily extended to allow a tolerable number of cases to persist. Nonetheless, this will have a large impact of the cost, and may lead to costly control measures such as lockdown to be implemented in waves.

Most studies on epidemic modelling and analysis focus on the relationship between epidemiological parameters and the effect of control measures [15, 17, 25], evaluation of the effectiveness of control measures in a specific outbreak [2, 4, 18], and the prediction of the trend of the ongoing epidemic [31, 85]. Compared with the existing researches, we proposed a more comprehensive model in which main epidemic control measures was embedded in, and considered the epidemic control, economic loss, and regional economic conditions and demographic status within the same framework. The objective of this paper is not limited to a specific epidemic, but takes the wave of COVID-19 epidemic in Xi'an City, Shaanxi Province as an example to discuss the prevention and control of emerging infectious diseases in general. In addition, we analyzed the impact of not only the characteristics of the infectious disease themselves but also demography and economics of the epidemic area on the control of the epidemic. Under the principle of minimizing the cost of control, we proposed the specific control plan which provides certain reference value for the response to possible outbreaks in the future.

## Supporting information

**S1 Fig. The number of daily new isolated cases in regions of Xi'an City, China.** Blue dots are observations and blue lines are model fitting values. Blue shaded areas are credible intervals.
(PDF)

**S2 Fig. The number of infected cases in the infectious period in regions of Xi'an City under the optimal control scheme.** The number of infected individuals $I_i(t)$ under the optimal control scheme shown in Fig 3 bottom panel. Black and green vertical dotted lines indicates the time when $I_i(t)$ and the number of new confirmed cases reaches zero. A region without a green line indicates that the number of new confirmed cases in that region is always zero.
(PDF)

**S1 Appendix. Supplement to methods and technical details of this paper.** This Appendix provides details of numerical simulation algorithm of continuous time Markov chain, details of the simulated annealing algorithm that we used to solve the optimal control problem, and supplementary illustrasions and simulation to some results in the main text.
(PDF)

## Author Contributions

**Conceptualization:** Fan Xia, Yanni Xiao.

**Data curation:** Fan Xia.

**Formal analysis:** Fan Xia.

**Funding acquisition:** Yanni Xiao, Junling Ma.

**Investigation:** Fan Xia, Yanni Xiao, Junling Ma.

**Methodology:** Fan Xia, Yanni Xiao, Junling Ma.

**Project administration:** Yanni Xiao, Junling Ma.

**Supervision:** Yanni Xiao, Junling Ma.

**Validation:** Fan Xia, Yanni Xiao, Junling Ma.

**Visualization:** Fan Xia.

**Writing – original draft:** Fan Xia.

**Writing – review & editing:** Fan Xia, Yanni Xiao, Junling Ma.

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
