## [Decision Letter · Decision Letter 0]

8 Apr 2024

Dear Dr. Ma,

Thank you very much for submitting your manuscript "The Optimal Spatially-Dependent Control Measures to Effectively and Economically Eliminate Emerging Infectious Diseases" for consideration at PLOS Computational Biology.

As with all papers reviewed by the journal, your manuscript was reviewed by members of the editorial board and by several independent reviewers. In light of the reviews (below this email), we would like to invite the resubmission of a significantly-revised version that takes into account the reviewers' comments.

Note that all three reviewers felt the manuscript needed to do a better job of citing the relevant literature and putting the research in context of recent advances in terms of data-driven metapopulation modeling and the impact of interventions during the COVID-19 pandemic. Some of the modeling assumptions also require better support. Please pay particular attention to addressing Reviewer 3's concerns, as we may not be able to accepted the revised manuscript if these concerns are not sufficiently addressed.

We cannot make any decision about publication until we have seen the revised manuscript and your response to the reviewers' comments. Your revised manuscript is also likely to be sent to reviewers for further evaluation.

Sincerely,

Virginia E. Pitzer, Sc.D.

Section Editor

PLOS Computational Biology

Virginia Pitzer

Section Editor

PLOS Computational Biology

Note that all three reviewers felt the manuscript needed to do a better job of citing the relevant literature and putting the research in context of recent advances in terms of data-driven metapopulation modeling and the impact of interventions during the COVID-19 pandemic. Some of the modeling assumptions also require better support. Please pay particular attention to addressing Reviewer 3's concerns, as we may not be able to accepted the revised manuscript if these concerns are not sufficiently addressed.

Reviewer's Responses to Questions

**Comments to the Authors:**

Reviewer #1: The paper exhibits commendable writing, delves into an intriguing topic, and demonstrates sound analytical and numerical analyses, rendering the work acceptable. However, I would like to offer some recommendations for improvement. My comments are outlined below:

1. The Introduction and Literature Review sections require a more detailed rewrite. The introduction should furnish a comprehensive summary of relevant prior works, with explicit references pinpointing the gaps in the existing literature.

2. The authors are advised to reconsider or redesign Figures 2 to 6. The red background in each figure appears aesthetically unpleasing and may hinder comprehension. A modification or redraw addressing the visual aspect would enhance the overall clarity of the figures.

To improve, authors can follow the following works.

Understanding the Impact of Vaccination and Self-Defense Measures on Epidemic Dynamics Using an Embedded Optimization and Evolutionary Game Theory Methodology, Vaccines, 11(9) (2023).

Impact of awareness in metapopulation epidemic model to suppress the infected individuals for different graphs, European Physical Journal B, 92, 199 (2019).

Evolutionary vaccination game approach in metapopulation migration model with information spreading on different graphs, Chaos, Solitons & Fractals 120, 41-55, (2019).

Reviewer #2: The study delves into optimizing the control measures for the Covid-19 Delta variant specifically within Xi’an City. It constructs a compartmental model that also incorporates the geographical distribution of the population. This model is then utilized within an optimal control framework to discern the ideal timing for implementing interventions. Given the dearth of models that integrate both the epidemic and its economic ramifications, this paper contributes to filling the gap. The model is quite complex but well formulated.

The Bayesian parameter estimation model entails the fitting of numerous parameters. A more extensive discussion regarding this process, along with an assessment of the outcomes in relation to existing models or literature, would enhance the comprehensiveness of the paper. While some parameters may not directly correlate with the specific context, it is still worthwhile to explore any relevant comparisons.

The authors acknowledge that the economic costs portrayed in the model are somewhat high-level. Given that this aspect constitutes a significant component of the paper's objectives, providing additional details on the sources of these costs, their units, and any adjustments made would be beneficial. What would be the implications of accounting for costs associated with hospitalisation and deaths?

Minor comments

Several figures could benefit from refinement to enhance clarity and ease of interpretation. For example:

• While most journal readers will likely be able to interpret chunks of the Figure 1 diagram, some may be difficult to interpret without more information, especially as the model is only described later. For example, a few of the states are not easy to interpret without finding where the main text refers to them. I suggest adding sufficient information to understand the diagram. Perhaps moving the diagram down to the model section also helps.

• Figure 5 C3 is difficult to interpret in places. Moving the legends, perhaps to the side or bottom, or adding a white background to the legend could be helpful.

• Fig 6 has a lot going on. Perhaps the figure can be simplified for quicker interpretation. For example, 6A could be separated into multiple pots. Perhaps there’s a clearer way to convey the message about R0, GDP and optimal control cost moving together.

Reviewer #3: This manuscript is about optimal and timely spatial deployment of non-pharmaceutical interventions (NPIs) at the early stages of an outbreak of an infectious disease to prevent it from establishing. The paper would be of great relevance four years ago, but the novelty at this point in time is questionable.

The paper is solid and, as far I as I can tell without access to the code and data, is technically sound.

However, I do not think this paper is suitable for PLOS Comp Bio for several reasons: 1) it is written for a technical audience and not put in a larger applicable context, 2) it covers a very narrow literature, ignoring vast areas of applications in epidemic modelling and 3) therefore missing the novelty and the relevance. What makes a disease easily controllable (Fraser et al Science), relevance of commuters for spreading influenza in the US (Viboud et al, Science), reducing contact rates as an intervention (papers like Prem et al that do that by using contact data) are all seminal papers in the field that would provide relevant context for this work, relate it to a broader set of data and applications, and clarify what exactly this paper adds in terms of novelty.

Authors make a lot of assumptions on flows between patches which are scaled by GDP and distance. The following sentence has a reference to Balcan et al 2009, but these authors use mobility data to capture the flows in their multiscale mobility networks, rather than distance and GDP. There is a vast amount of literature on flows between patches (from mobility data (e.g. cellphone data, and airline traffic data, google mobility), census data (e.g. using postcodes of home and workplaces to infer flows), traffic, work place data) so it is surprising not to see any reflection on that literature in the manuscript and link it with the assumptions about the flows here. Referring to a study that analyses suitability of using GDP in scaling of gravity models would be one way to clarify this.

There are several instances of inappropriate referencing of claims in the paper, for example, citing newspaper article (ref [21]) to support effectiveness of a strategy rather than peer-reviewed analysis of effectiveness of a strategy). “Since contact tracing is only effective if the outbreak is detected early before widespread community transmission [21], ([21] People’s Daily Online. Interpretation of the new policy of epidemic prevention in Shanghai; 2022. Available from: http://health.people.com.cn/n1/2022/0319/c14739-32378995.html.) Instead, the validity of this statement should be supported by a published peer-reviewed paper or shown with own work presented in this manuscript.

There are several technical issues that need clarifying/justifying, for example:

a) lockdown is assumed to have different impact in different locations – is this modelling adherence/ compliance/ or that different locations have different proportions of e.g. keyworkers whose work and commute patterns will be different form non-key workforce? Please elaborate this.

b) Contact trancing is inherently an individual-based process and not straightforward to capture in population-level, compartmental models. Why is a logistic decay function a suitable one for modelling rates of isolation of exposed and infectious individuals? The only reference in this paragraph (lines 147-153) is a newspaper article, which I doubt will inspire readers of Plos Comp Bio with confidence.

c) Whole sections of paper (e.g. Estimation section, Optimal control section) have no references to other literature (e.g. MCMCs, Bayesian inference, modelling over-dispersion, defining a cost function and specifying an optimal control problem, using methods for minimising the cost or utility function should all refer to the relevant literature).

For all of these reasons I think this manuscript is not of sufficient novelty and interest to PLOS Comp Bio audience but I look forward to seeing it published in a specialised, technical journal.

**Have the authors made all data and (if applicable) computational code underlying the findings in their manuscript fully available?**

Reviewer #1: None

Reviewer #2: None

Reviewer #3: **No: **The authors say this will be made available should the paper be accepted, but as far as I am aware it hasn't been made available to reviewers. Or it is not easily accessible.

PLOS authors have the option to publish the peer review history of their article (what does this mean?). If published, this will include your full peer review and any attached files.

Reviewer #1: No

Reviewer #2: No

Reviewer #3: No
---

## [Decision Letter · Decision Letter 1]

17 Sep 2024

Dear Dr. Ma,

We are pleased to inform you that your manuscript 'The Optimal Spatially-Dependent Control Measures to Effectively and Economically Eliminate Emerging Infectious Diseases' has been provisionally accepted for publication in PLOS Computational Biology.

Best regards,

Feng Fu

Academic Editor

PLOS Computational Biology

Virginia Pitzer

Section Editor

PLOS Computational Biology

Reviewer's Responses to Questions

**Comments to the Authors:**

Reviewer #1: Thank you for your efforts.

**Have the authors made all data and (if applicable) computational code underlying the findings in their manuscript fully available?**

Reviewer #1: None

PLOS authors have the option to publish the peer review history of their article (what does this mean?). If published, this will include your full peer review and any attached files.

Reviewer #1: No

---

## [Editor Report · Acceptance letter]

23 Sep 2024

PCOMPBIOL-D-23-01902R1 

The Optimal Spatially-Dependent Control Measures to Effectively and Economically Eliminate Emerging Infectious Diseases

Dear Dr Ma,

I am pleased to inform you that your manuscript has been formally accepted for publication in PLOS Computational Biology. Your manuscript is now with our production department and you will be notified of the publication date in due course.

With kind regards,

Zsofia Freund
